# Indigenous Australians’ Experiences of Cancer Care: A Narrative Literature Review

**DOI:** 10.3390/ijerph192416947

**Published:** 2022-12-16

**Authors:** Saira Sanjida, Gail Garvey, James Ward, Roxanne Bainbridge, Anthony Shakeshaft, Stephanie Hadikusumo, Carmel Nelson, Prabasha Thilakaratne, Xiang-Yu Hou

**Affiliations:** 1Poche Centre for Indigenous Health, The University of Queensland, Brisbane, QLD 4072, Australia; 2School of Public Health, University of Queensland, Brisbane, QLD 4072, Australia; 3Institute of Urban Indigenous Health, Windsor, Brisbane, QLD 4030, Australia; 4Royal Brisbane and Women’s Hospital, Herston, Brisbane, QLD 4029, Australia

**Keywords:** indigenous people, aboriginal, cancer care, patients’ experiences, health communication, cultural safety, healthcare service, primary healthcare, hospital care, transitional care

## Abstract

To provide the latest evidence for future research and practice, this study critically reviewed Indigenous peoples’ cancer care experiences in the Australian healthcare system from the patient’s point of view. After searching PubMed, CINAHL and Scopus databases, twenty-three qualitative studies were included in this review. The inductive approach was used for analysing qualitative data on cancer care experience in primary, tertiary and transitional care between systems. Three main themes were found in healthcare services from Indigenous cancer care experiences: communication, cultural safety, and access to services. Communication was an important theme for all healthcare systems, including language and literacy, understanding of cancer care pathways and hospital environment, and lack of information. Cultural safety was related to trust in the system, privacy, and racism. Access to health services was the main concern in transitional care between healthcare systems. While some challenges will need long-term and collective efforts, such as institutional racism as a downstream effect of colonisation, cultural training for healthcare providers and increasing the volume of the Indigenous workforce, such as Indigenous Liaison Officers or Indigenous Care Coordinators, could effectively address this inequity issue for Indigenous people with cancer in Australia in a timely manner.

## 1. Introduction

Cancer is a major burden of disease worldwide [1]. In Australia, the burden is higher in Aboriginal and Torres Strait Islander people (hereafter respectfully referred to as Indigenous people) than non-Indigenous people. It is the leading cause of death for Indigenous peoples [2,3]. Recognising the higher burden of cancer among Indigenous peoples, official guidelines and strategies have been developed, such as the Australian Government and Cancer Australia’s seven steps of optimal care pathway for health practitioners and service planners for Indigenous people [4]. In Australia’s primary healthcare network, patient-healthcare professional interaction [5,6] using ‘culturally competent care, effective communication, coordination, collaboration, and timely information exchange’ (page 3) [7] are the key components of quality cancer care outcomes.

Research evidence notes a range of barriers to delivering the best possible quality cancer care to Indigenous people. Barriers include navigating complex cancer treatment and management pathways (often associated with late diagnosis and multiple comorbidities) [8,9], inadequate provision of long-term and continuous care to survivors [10], and patients’ distrust of the healthcare system [11]. Delivery of health services in rural/remote areas is another barrier due to the lack of access to services (such as specialist care) and the workforce shortage compared to services available in urban or larger regional cities [12]. The social and cultural determinants of health are also documented as factors that can negatively impact Indigenous peoples’ access to cancer care at all stages [13,14].

Research focusing on healthcare professionals has also highlighted barriers to cancer outcomes for Indigenous peoples, including interactions between patients and healthcare professionals [5,6] and cultural sensitivity among healthcare providers [15]. A study of Indigenous cancer care providers highlighted essential cancer care components: culturally competent and responsive care; providing psychological support; ascertaining and responding to patient needs; delivery of practical assistance; and advocating for Indigenous health [16]. A systematic review of carers’ experiences identified five major gaps in cancer care pathways that need attention: information, support, communication, balancing roles and emotions, and culturally unsafe healthcare systems and settings (page 10) [17].

The Research Alliance for Urban Goori Health (RAUGH) is a tripartite association between the Poche Centre for Indigenous Health at the University of Queensland, Metro North Hospital and Health Service (MNHHS), and the Institute for Urban Indigenous Health (IUIH). The aims of our multidisciplinary team with leading Indigenous researchers are to close the gap in life expectancy and achieve health equality for urban First Peoples in greater Brisbane North through applied research in priority areas in the whole health system, including primary, tertiary, and transitional care. Improving cancer care for Indigenous people with cancer has been identified as one of the priorities by MNHHS and IUIH.

To date, limited research has focused on Indigenous cancer patients’ perspectives and experiences of cancer care. Indigenous people’s real-world cancer care experiences, such as how they faced their cancer challenges by going through multiple healthcare services, are largely absent but needed. Understanding Indigenous cancer patients’ care journeys and perspectives must complement clinical research to improve patient outcomes and efficiencies. It is also required for decision-making processes to avoid health inequity in healthcare policy [18,19]. We, the RAUGH team, conducted a narrative literature review to show current evidence about the experiences of Indigenous people with cancer care services in Australia’s primary and tertiary healthcare systems. The findings will guide future research and clinical practice in Australia and other countries.

The research question for this literature review is: What are the experiences among Indigenous people with cancer when they go through different sectors of the healthcare system in Australia? We hypothesise that their experiences would include both positive and negative aspects, which may include racism and cultural safety issues.

## 2. Methods

### 2.1. Search Strategy

Three electronic databases, PubMed, CINAHL, and Scopus, were searched to identify qualitative studies of Indigenous people’s cancer care experiences in Australia, from diagnosis to survivorship care. The healthcare services in cancer included both primary and tertiary systems. Studies were included if they reported Indigenous patients’ healthcare service experiences, their expectations from healthcare service providers, and gaps between the systems in cancer care. The database searches were limited to studies published in English in peer-reviewed journals until June 2022. After consultation with an experienced librarian, the main keywords and Medical Subject Headings (MeSH) search terms were ‘Indigenous people’ AND ‘cancer’ AND ‘healthcare service’ AND ‘Australia’ AND ‘community’ AND ‘health service’. The relevant reference lists of previously published reviews were searched manually. The details of the search strategies with keywords and databases are presented in Appendix A.

### 2.2. Study Selection

Full-text, peer-reviewed journals published in English were included. Eligible studies included adult Indigenous people (18 years or older) who received cancer care or services (including assessment, treatment, follow-up care and survivorship) and carers or family members as the representatives of cancer survivors. There were no restrictions on cancer type, stage, or treatment. Only qualitative studies using interviews for data collection or secondary qualitative data were selected for the literature review. Two or more studies that presented the same participant experiences from different perspectives of the healthcare services (including primary, tertiary or transitions between them) were also included in this review to explore the range of cancer care services. The exclusion criteria were studies with (1) Indigenous people aged 18 years or less (those who were under 18 years but became adult cancer survivors were included), (2) healthcare professionals or related service providers or Aboriginal Liaison Officers (ALOs), (3) a focus on cancer screening only, (4) palliative care treatment only, (5) a focus on exploring knowledge about cancer only, or (6) reviews, book chapters, and abstracts published in conferences, editorial materials, correspondence, letters, case reports, newspaper materials, or dissertations.

Two authors (S.S. and X.-Y.H.) were involved in the study selection procedure, including the preliminary search, systematic database searches, title and abstract screening, and full-text screening. The EndNote database was used to manage the searched/found articles. After removing duplication, titles and abstracts were screened after considering the inclusion and exclusion criteria of the studies. Two authors (S.S. and X.-Y.H.) assessed the full-text articles, and the discussion resolved any disagreements.

### 2.3. Data Extraction

The author (S.S.) randomly selected five included studies to prepare a preliminary data extraction format in the Excel file. The data extraction procedure was conducted in four steps. First, one author (S.S.) read the studies and extracted the study characteristics (including title, year of publication, study design, main aim, timeline, and data collection methods) and participants’ characteristics (including sample size, age, site in Australia, community, socio-economic status, type of cancer and their treatment, type of services received with their positive or negative outcomes, main findings, expectations from the services and service gaps). Second, an inductive approach was used to qualitatively evaluate individual studies’ themes and quotations [20]. Specific segments of Indigenous people’s cancer care service experience were extracted and categorised with labels to understand the service gaps in primary and tertiary healthcare systems. Carers’ cancer care experiences closely related to Indigenous peoples’ perspectives were extracted, and no other details (such as sample size, gender, etc.) of carers’ information were included in this review. Third, authors re-read the texts and labels to reduce the overlap and redundancy among titles. Fourth, the second author (X.-Y.H.) randomly selected 10% of the total articles and conducted data extraction following the previous steps.

## 3. Results

### 3.1. Database Search Results

In total, 768 articles were found in the database searches: PubMed (*n* = 494), Scopus (*n* = 110), and CINAHL (*n* = 164) (Appendix A). References of reviews and research studies were searched manually (*n* = 133). After removing duplicate studies, the titles and abstracts of 543 studies were screened, with 59 studies identified for full-text screening. Twenty-three qualitative studies [18,19,21,22,23,24,25,26,27,28,29,30,31,32,33,34,35,36,37,38,39,40,41] were included in the review after excluding quantitative studies (*n* = 16), studies using the same dataset with different aims (*n* = 3), studies not focused on Indigenous people’s experience with cancer (*n* = 7), and studies not focused on Indigenous people with cancer (*n* = 10).

### 3.2. Study and Participants’ Characteristics

Table 1 presents the study and patient characteristics of the articles included in this literature review. A total of 244 Indigenous people with cancer were interviewed in 21/23 studies (Table 1). Two studies [27,31] did not specify the number of Indigenous people with cancer included in the studies. At least 145 participants (60%) were from regional/rural or disadvantaged areas. Ten studies had the same participants who shared their experiences in different healthcare service areas living in Northern Territory (*n* = 3) [24,25,26], New South Wales (*n* = 2) [29,38], and Western Australia (*n* = 5) [33,34,35,36,39]. Most of the studies (*n* = 21) conducted state-wide research; only two were conducted at the national level [21,41]. Six studies specifically focused on Indigenous women’s health (*n* = 6/23) and diagnosed with gynaecological (*n* = 5/6) and breast (*n* = 1/6) cancer.

### 3.3. Main Findings

Table 2 presented three major themes identified from Indigenous people’s experience in cancer care pathways: communication, cultural safety, and access to services. Different sub-themes were collated and highlighted the specific issues among patients’ healthcare service experiences in Australia’s primary and tertiary healthcare and the transition between healthcare services.

It needs to be noted that these findings are directly from Indigenous people with cancer. We gratefully acknowledge their time and effort in contributing to the research findings which build the body of scientific evidence. The content of the challenges presented here only aims to provide insights to improve future cancer care in the health system, not to blame the people who already had negative experiences.

#### 3.3.1. Theme 1: Communication

Communication was one of the main themes identified from Indigenous people’s experiences in healthcare systems, including primary, tertiary, and transitional care between the services (Table 2). Four related sub-themes and key issues were found in communication, which caused difficulties in taking up healthcare services in the systems.

Language and literacy: Language was the most important sub-theme in communication because the use of unfamiliar medical terms and procedures of cancer treatment increased the complicacy of the entire treatment pathway [25]. Sometimes this complicacy started with hospital directories. For instance, using hospital maps to find the location of treatment areas inside the hospital was challenging because some Indigenous patients could not read the maps’ signs [35,39]. Being illiterate or old, or unfamiliar with medical terminology also made Indigenous people struggle with electronic communication, such as using the touchscreen or online surveys [21]. Patients also had difficulties understanding their medication from different providers [28], confusing terminology and written documents [39], complex discharge reports in written statements due to illiteracy [28], and understanding the entire treatment procedure [25]. The consequences of language barriers and poor coordination included missing appointments [23,28].

Health professionals’ lack of interest in seeking family assistance to communicate information for Indigenous patients with literacy difficulties was also found in the treatment decision-making process. Even during the signing of official documents, there was no proper communication on the delivery or understanding of the treatment [31].

Understanding of the hospital environment: After travelling hundreds of kilometres for several days (such as travelling via plane, taxi or bus), Indigenous patients faced entirely different experiences staying at hotels or other accommodations, understanding the transport system, cost of food, and so on [35]. The impressions of the hospital environment for Indigenous people with cancer were “Alienating” and “Isolated” (page 576) compared to their country (residence) [36]. Patients from rural or remote areas felt different from those from cities regarding their understanding of the whole hospital environment, including healthcare staff or professionals, treatment facilities, and visitor numbers [32,35]. Some patients feared and were anxious about the hospital environment, such as lifts, surgery procedures, recovery time, visiting specialists, medical tools, and sterile rooms. [24].

Understanding of cancer care pathways: Navigating a complex healthcare system for cancer care pathways is a significant challenge for patients and carers [39]. Even understanding hospital appointment schedules at the beginning of cancer care pathways was complicated for Indigenous people when no support was available [18,19,35]. In some instances, the cancer care pathway was challenging for Indigenous women diagnosed with cancer at an advanced stage, who had to rush to start treatment without a proper explanation from a General Practitioner (GP) [27]. Ultimately, it caused negative feelings about not managing the treatment appropriately [27]. According to Indigenous people’s experience, the whole hospital system was “cold, indifferent and unwilling to tolerate when approached to the system” (page 3) [39]. Even a decade later, Indigenous people still felt a lack of understanding of cancer care pathways in the hospital system and needed to take time in decision-making in cancer treatment [23,29].

Lack of information: Not knowing where and how to get health or cancer-related information for cancer treatment is another disappointment for Indigenous people, specifically after diagnosis at an advanced stage and the urgent start of therapy [21,27,35]. Dissatisfaction was arisen among Indigenous people with gynaecological cancer after moving to the hospital for cancer treatment; they felt they did not receive enough cancer information, even after receiving treatment from a metropolitan hospital [23,25,27,37]. Patients also shared their experiences about a lack of sufficient and appropriate information about their discharge reports, including treatment and medication side effects [31], follow-up appointments, upcoming disease conditions [32], and access to support [19]. This left many patients “bewildered and stressed” about diagnosis, treatment, and follow-up plans (page 6) [23]. In some cases, carers of Indigenous people with cancer felt they received limited information about cancer as a carer because usually, the hospital provides the information targeted to the patients rather than carers [19]. If the carer or family member received the report, they might give proper information support to the patients [36].

#### 3.3.2. Theme 2: Cultural Safety

Indigenous people faced some cultural safety issues in their cancer care pathways, especially in primary and tertiary healthcare. These cultural safety issues include trust in the system, privacy, racism, patient-healthcare provider relationship, and removal of body parts (Table 2).

Trust in the system: GPs are the main healthcare professionals providing quality care services in the primary healthcare system. Unfortunately, some Indigenous people believed that GPs’ lack of awareness of cancer diagnosis delayed subsequent treatment procedures [23,29,36,37]. Even after informing of symptoms, poor continuity of care by healthcare professionals in primary care impacted the patient-healthcare relationship and caused a lack of trust in the system; it resulted in being seen by multiple GPs (shop around) and delayed the cancer treatment pathways [29,34,37]. Indigenous people felt they needed culturally-appropriate personal services such as an Indigenous Support Officer (ISO or ALO) at the beginning of the treatment programs, so they could understand the whole procedure before leaving their country [35].

Lack of trust was also found in tertiary care. Sometimes Indigenous people lost their confidence and trust in healthcare professionals and cancer treatment due to alienating tertiary-care environments [23]. In some instances, trust was related to not fulfilling the high expectation from the healthcare system, such as the full recovery from cancer. Indigenous patients’ confidence level in the system deteriorated if they saw someone’s death or were not recovering from cancer [31].

Privacy: Cancer is a personal issue for Indigenous women as they consider it “women’s business” (p-2772); they don’t even discuss it with other family members [25,31]. Some women living in regional or remote areas preferred to be examined by a female GP or someone they knew; they were too shy to talk about gynaecological problems to a GP and chose to speak in front of a female family member or husband [25,41]; they want to maintain privacy and are reluctant to receive any support for relocation for treatment [23]; they didn’t want any physical touch by healthcare professionals in the hospital. Some women preferred female staff in every aspect of the hospital system [41]. Some patients felt the invasion of privacy due to medical students, other ward staff, and even medical professionals rounding in the ward [36].

Racism: Indigenous people with cancer or affected by cancer expressed negative feelings about racism or disrespectful behaviour of hospital staff [21,27,30,37,40]. In some cases, Indigenous people felt disrespected due to the lack of records about carer information in the system or healthcare professionals’ ignorance to recognise the carer of patients [19]. Some women from rural areas chose not to take treatment or delay the treatment due to racism and discomforting experiences with healthcare professionals [36] or preferred female staff due to “possible acts of racism than issues of sex” (p. 102) [41].

Some Indigenous people were dissatisfied with their healthcare professionals regarding their lack of warm interaction [39], communication in a culturally unsafe way [18], unsympathetic delivery of bad news [23], not maintaining patients’ privacy [19,31], absence of Indigenous culture or unable to practise cultural ceremonies [21], or importance of people’s cultural and family support in cancer patient’s treatment and recovery [31,39]. The removal of hair or body parts due to surgery was also a cultural safety issue for Indigenous people [26,27], considering the removal a violation of culture because body parts should return safely to their country.

#### 3.3.3. Theme 3: Access to Services

The significant issues in access to services were mainly found in transitional care between primary and tertiary healthcare services and vice versa, such as long waiting times, availability of follow-up services and coordination between services (Table 2).

Long waiting time: There were identified gaps between primary and tertiary healthcare services in Indigenous people living in regional or remote areas due to long waiting times. For example, there were reported significant delays in cancer diagnosis and treatment due to long waiting times for specialist referrals from GPs and poor communication from public hospitals [23,37,40]. In addition to the referral and communication delays, visiting specialists over long distances was another problem for Indigenous people in remote areas. Seeing specialists in larger rural towns was difficult, and when they finally arrived, they waited a long time during the visit [24,41].

Availability of follow-up services: Research findings showed that Indigenous people with cancer were worried about their ongoing needs for follow-up care [22], including follow-up appointments, medication access, side effects, adverse effects [31,32], follow-up care [28], and lack of coordinated care or follow-up from allied health services (such as physiotherapy and counselling) in rural areas [30,35]. Unfortunately, the unavailability of services due to vacant positions or the absence of follow-up care was a significant problem [18]. Staff turnover in hospitals and explaining the whole history of medical records at follow-up care also led to non-compliance with follow-up treatment [35].

Indigenous people with gynaecological cancer [23] reported being confused and anxious about their treatment plan and prognosis, particularly regarding the transition from tertiary to primary healthcare [21]. There were no proper follow-up instructions on postoperative complications such as pain, wound management, and side effects [25], no extra care for long-distance travel for women with gynaecological cancer surgery [23], and minimal resources were available in the community services in rural and remote areas, which caused inconsistent follow-up care, such as regular appointments [24].

Coordination between services: Some patients reported a lack of coordination with a GP between tertiary and primary services [28,36]. Maintenance of a healthy lifestyle is essential during follow-up cancer care and requires guidance from allied health professionals such as dietitians and psychologists. Unfortunately, Indigenous people reported that they received the appointments at night, which was then challenging for them to maintain appointments [23]. Coordinating between primary care services and tertiary hospitals to make appointments was a significant problem. The staff who created the hospital appointment did not consider the patients living status (rural or remote), availability or cost of travel, or involvement in primary care services [28,36].

## 4. Discussion

This literature review was the first study that critically presented Indigenous people’s cancer care experiences in Australia’s primary and tertiary healthcare services. As we anticipated in the hypothesis, with very few positive and encouraging findings about Indigenous Australians’ experiences in the health system, most of the studies reported areas for improvement from their research participants. Some key issues, including communication, cultural safety, and access to services, have been identified in Indigenous people’s experiences, which can help to improve the healthcare service gaps between Indigenous and non-Indigenous people in Australia.

For Indigenous people with cancer, it is obvious that language barriers and low health literacy play a vital role in reducing their understanding of the cancer care pathways and consequently reducing their ability to join the decision-making process in their cancer journey [36]. This barrier leads to miscommunication between patients and healthcare professionals regarding patients’ cancer care pathways [28,37]. The worst-case scenario of this communication gap was for these Indigenous patients to leave the hospital without cancer treatment, resulting in non-compliance with treatment [25,30]. Unfortunately, a systematic literature search found inadequate evidence of health literacy measures in Indigenous people to understand cancer care outcomes. It warrants further research to quantify the current health literacy status [42].

The healthcare service should be tailored for all walks of life, including people with low health literacy. Therefore, the responsibility to improve the communication between Indigenous patients and healthcare professionals should mainly be with healthcare professionals. At the same time, we acknowledge that improving Indigenous people’s levels of literacy, including health literacy, will further improve their quality of life.

It is more pleasing to see research demonstrating effective measures to improve communication with Indigenous people with cancer. Simple interventions such as booklets and pamphlets made Indigenous people feel that they were informed about using helpful resources to get cancer-related information [23]. Complex interventions may involve the Indigenous workforce in the health system. Supportive care staff, such as ALOs or Care Coordinators, helped patients to understand health systems and to communicate between patients and healthcare professionals [18,21]. The roles of ALOs include logistic support, helping patients find and access the available services in transitional care for cancer treatment [19,24,27,30], supporting patients with the required services, providing treatment information [30], care coordination and family arrangements [35]. As most of the service assistants were from non-medical backgrounds, they could not help with information related to cancer and treatment [18,43]. Therefore, specialised training for the Indigenous support staff will empower them to provide a higher quality of support to Indigenous people with cancer, from access to cancer care services to understanding treatment procedures, medication management, and discharge summaries.

In addition to the communication barrier, the research reported that Indigenous people do not trust mainstream health services, which could be due to the limitations in the health system itself, historical reasons, and the intergenerational traumatic experience of colonisation. This lack of trust worsened when Indigenous people were treated without respect. On the other hand, healthcare professionals’ positive behaviour towards Indigenous people had improved patients’ trust in the health professionals for diagnosis and cancer treatment [41], despite the small number (two out of 24 women) in Willis et al. 2011 study. Disrespectful behaviours from health professionals towards Indigenous people with cancer could be due to a wide range of reasons, including reasons within the health system, such as institutional racism and lack of respect for Indigenous culture. This was evident in the findings about Indigenous cancer patients’ privacy and their preference in choosing health professionals.

Regarding the recommendations to improve cultural safety for Indigenous people with cancer, cancer care services have involved multidisciplinary patient-centred teams and continuously tried to deliver culturally safe and accessible services, including those in hospitals [7,38,44]. The published recommendations include providing more positive feedback to patients, particularly in end-of-life care [18], giving more time for patients to discuss the cancer journey and required resources for patients’ survivorship [21,23], sharing cancer care plans directly with young survivors rather than their parents [32], improving patients’ language and literacy, cultural training for staff, and staff involvement [45].

Delivering cultural safety training to hospitals or community services staff could be essential to improving the cultural safety of Indigenous people with cancer. A recent study found that around half of the healthcare professionals had attended cultural safety training to treat Indigenous people to better implement the optimal care of cancer pathways in primary healthcare [15]. Even a single workshop improved healthcare professionals’ confidence concerning cultural differences [46] and may improve healthcare professionals’ understanding of maintaining the privacy of cancer care [29].

Regarding access to cancer care services, having a long-term relationship with local GPs seems to be important in contributing to Indigenous patients’ positive experiences, especially when they believed that their GP was fully informed to provide the services such as cancer follow-up care, management of multiple comorbidities, and understanding of cancer [28,32]. Consequently, some Indigenous people were also pleased with the healthcare providers and other staff support throughout the treatment procedure [32,37]. Access to local primary healthcare services and engagement with the staff helped Indigenous people with cancer, supported the patients to keep the treatments in the hospital, avoid discharge against medical advice [38], and even helped the Indigenous people living in rural and regional areas with better chemotherapy outcomes, and increased patients’ comfort within the services [32]. Therefore, patients were happy with the journey between rural/remote areas to the tertiary hospital [35] and could understand the cancer treatment and its process [25].

Despite a small number of publications, there are other positive experiences from Indigenous people with cancer in Australia. Some Indigenous patients showed realistic expectations, acceptance of cancer after diagnosis, expected quick clinical treatment options (rather than emotional feelings support), responded positively to the treatment and service received [22], coped well with a strong desire to live, accepted their physical and functional capabilities [29], felt very positive about health check-ups and treatment [25,27] and access to services during discharge including their understanding of prognosis, side effects and overall follow-up [28]. Some Indigenous patients asked for more contact and time during follow-up care [23,36], which is a patient-initiated activity. In addition, patients raised the need for emotional and psychological support for their carers to maintain their psychological well-being during the cancer care pathways [19,21,35]. With a strong family culture in the Indigenous community, it is not surprising that patients hope to get help or guidance for their family members about the diagnosis and prognosis of cancer type and services available or required during cancer survivorship [19].

Although this narrative literature review explored real-world Indigenous people’s cancer care service experiences, several limitations in the existing literature warrant more research. The participant number in the seven studies was low (range of 3–10 participants), which might represent only a small community section [22,23,29,32,37,38]. The studies may have sample selection bias [30], resulting in biased results or service delivery systems. Using the same dataset from a different research perspective may repeat the findings. Unfortunately, the experience in the broader Indigenous community with different cultural backgrounds and language diversity is unknown. This study excluded Indigenous cancer survivors under 18 years, which might not express real-world cancer care experiences in primary and tertiary care. Young survivors may be under the control of caregivers, and a worldwide systematic literature review of caregivers’ cancer care experience has already been published [17].

The interviews were conducted after the completion of the treatment or recovery phase. Limited research on cancer service experience was found during cancer treatment, including multiple comorbidities, cancer stage, cancer treatment outcomes (such as surgery or chemotherapies) or peer-support services received in a hospital or throughout survivorship. Most of the studies were conducted state-wide, so there will be some variation in the delivery of healthcare services in different states in Australia, especially in the tertiary care sectors. Most importantly, recent research on measuring healthcare experience tools showed limitations in covering Indigenous people’s cultural safety [47], potentially resulting in limited research outcomes on cultural safety issues in the current literature.

The findings in this literature review may also be transferable to other health systems. For example, the findings could be relevant to the health system in Europe, especially the unpredictable volume of immigration from other countries such as Africa and Syria. However, we should not assume similar characteristics between immigrants in Europe and the First Nations in Australia. It would be interesting to see the research findings on this topic from colleagues in Europe.

In this modern era of patient-centred healthcare and the politically committed goal of “Close the Gap” in Australia, Indigenous people’s subjective experiences are as important as their clinical characteristics in the joint journey of cancer care with health professionals. Our health systems need to care for the “person”, not just their “cancer”.

## 5. Conclusions

This review critically reviewed the gaps (communication, cultural safety, and access to care) in cancer care services in the primary and tertiary systems and transitional care for Indigenous people with cancer in Australia. At the strategic level, some challenges, such as institutional racism as a downstream effect of colonisation, warrant long-term and collective efforts of the healthcare system. At the operational level, cultural training for healthcare providers to improve cultural and communication competency and increasing the Indigenous workforce, such as Indigenous liaison officers or care coordinators, could effectively address this unacceptable inequity issue for Indigenous people with cancer in Australia and other countries.

## Figures and Tables

**Table 1 ijerph-19-16947-t001:** Study and patients’ characteristics.

Author and Year	Timeline	Indigenous People *	State	Remoteness	Type of Cancer	Main Research Interest
Anderson, 2021 [18]	2017	75	NT	65/75 relocated for treatment	All	Accessibility of cancer treatment services
Bell, 2021 [19]	2019	15	QLD	Metropolitan (*n* = 12) Regional (*n* = 3)	All	Support needs
Green, 2018 [21]	2016	24	VIC, NT, NSW	Metropolitan (*n* = 5) Regional (*n* = 10) Remote (*n* = 9)	All	Experience in cancer care
Lyford, 2018 [22]	-	3	WA	Regional (*n* = 3)	Rare type	Underrepresentation of regional services
Marcusson-Rababi, 2019 [23]	2017	8	QLD	Disadvantaged areas (*n* = 7)	Gynaecological	Women’s experience of cancer care
McGrath, 2013a [24]	2011	12	NT	Remote (*n* = 12)	Vulvar	Experience of relocation for cancer treatment
McGrath, 2013b [25]						Treatment of cancer
McGrath, 2015 [26]						Treatment of cancer
McMichael, 2000 [27]	1998–1999	-	QLD	Included urban, rural and remote areas participants	Breast	Treatment and post-treatment care and support of cancer.
Meiklejohn, 2017 [28]	2015–2016	21	QLD	Major city (*n* = 12) Regional (*n* = 4) Remote (*n* = 5)	All	Follow-up cancer care and management
Treloar, 2013 [40]	2008–2011	22	NSW	-	NR	Cancer care process
Newman, 2017 [29]		6				Engaging with a cancer treatment
Reilly, 2018 [30]	2015–2016	29	SA	Urban (*n* = 11) Regional (*n* = 3) Remote (*n* = 18)	All	Cancer and care coordination
Ristevski, 2020 [31]	-	3	VIC	Regional (*n* = 3)	NR	Survivorship care models
Sariman, 2022 [32]	-	10	QLD	Regional (*n* = 8) Remote (*n* = 2)	NR	Experience of cancer
Shahid, 2009a [34]	2006–2007	14	WA	Mostly rural and remote	NR	Access to cancer services
Shahid, 2009b [36]						Participation in cancer care
Shahid, 2010 [33]						Cancer and cancer services
Shahid, 2011 [35]						Access to cancer care services
Thompson, 2011 [39]						Participation in cancer treatment
Taylor, 2020 [38]	2015–2018	5	WA	Urban (*n* = 1), Regional (*n* = 2), Remote (*n* = 1)	NR	Respect for healthcare
Taylor, 2021 [37]						Experience of cancer services
Willis, 2011 [41]	2008	3	NSW, SA, VIC, NT	Rural (*n* = 3)	Gynaecological	Expectations of clinical care

* with cancer participants included in the studies. NR: Not reported.

**Table 2 ijerph-19-16947-t002:** Key themes and issues found in Indigenous peoples’ cancer care experiences.

**(A) Communication**	
**Sub-Themes**	**Specific Issues That Cause Difficulty (Reference)**
Language and literacy	Unfamiliar medical terms and procedures [25] Reading location maps and signs [35,39] Confusing information about medications [28] Electronic communication [21] Complex discharge reports [28] Unfamiliar medical terminology [26] Missing appointments [23,28] Health professionals’ lack of interest in seeking family assistance [31]
Understanding of the hospital environment	Felt “Alienating” and “Isolated” in the hospital [36] Unfamiliar hospital environments such as healthcare staff or professionals, treatment facility, and visitor numbers [32,35] Feared and anxious about hospital facilities [24]
Understanding of cancer care pathways	Lack of understanding of the pathway from primary to tertiary care [21,27,35] Lack of explanation from GP [27]
Lack of information	Lack of understanding of treatment in tertiary care [21,27,35] Insufficient information in the discharge report regarding medication [31], follow-up appointment [32], disease condition [32] or access to support [19] Limited information for carer [19]
**(B) Cultural safety**	
**Sub-Themes**	**Specific Issues That Cause Difficulty (Reference)**
Trust in the system	GPs’ lack of awareness of cancer diagnosis delayed subsequent treatment procedures [23,29,36,37] Seeing multiple GPs and delayed cancer treatment [29,34,37] Lack of confidence due to the alienating environment [23] High expectations from tertiary care, such as complete recovery [31]
Privacy	Lack of interest in ‘women’s business’ [24,31] Preference of examiner for health checks in primary and tertiary care [25,41] Reluctant to receive treatment due to privacy [23] Hospital staff visits [36]
Racism	Racism behaviour of hospital staff [21,27,30,36,37,40] Disrespectful due to lack of carer information [19] Lack of warm interaction in the hospital [39] Communication in a culturally unsafe way [18] Unsympathetic delivery of bad news [23] Not maintaining patients’ privacy [19,31] Absence of Indigenous culture or unable to practice cultural ceremonies [21] Cultural and family support in treatment and recovery [31,39] Removal of hair and body parts due to surgery [26,27]
**(C) Access to services.**	
**Sub-Themes**	**Specific Issues That Cause Difficulty (Reference)**
Long waiting time	Long waiting times for specialist referrals from GPs [23,37,40] Poor communication from public hospitals [23,37,40] Long waiting time for visiting specialists [24,41]
Availability of services at follow-up	Treatment prognosis from tertiary to primary care [21] Ongoing needs for follow-up care such as follow-up appointments, medication access, side effects, and adverse effects [22,28,31,32] Lack of follow-up instruction on postoperative complications or side effects [23] Travelling precautions from hospital to home [23] Lack of coordination in rural areas [30,35] Community services in rural and remote areas [24] Maintenance of follow-up appointments in the hospitals [24]
Coordination between services	Coordination problem between tertiary to primary services (GP) [28,36] Unplanned appointment schedules from allied health services and hospital staff [23,36,38]

## Data Availability

Data is available upon reasonable request.

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
