# Peer review of "Indigenous Australians’ Experiences of Cancer Care: A Narrative Literature Review"

_ijerph, 2022, doi:10.3390/ijerph192416947_

Round 1

Reviewer 1 Report

Thank you for the opportunity to review the manuscript “Indigenous Australians’ experiences of cancer care: A narrative literature review” (ijerph-2063110).

The authors submitted a paper about Indigenous peoples’ cancer care experiences in the Australian healthcare system from the patients' point of view.

A clear research question and hypotheses are missing at the end of the introduction.

The research question and hypotheses must then be taken up again in the discussion and addressed with the available literature.

This requires a fundamental revision of the paper.

The authors should add a table with the most important findings of the review.

Strategic and operational recommendations for action would be important to tackle the problem. (Conclusion)

The problems are also transferable to other health systems in Europe due to the explosive immigration of Africans and Syrians.

Author Response

Reviewers’ Comments

Authors’ Responses

The authors submitted a paper about Indigenous peoples’ cancer care experiences in the Australian healthcare system from the patients' point of view.

A clear research question and hypotheses are missing at the end of the introduction.

Thank you for the feedback.

At the end of the Introduction, we managed to add a section about research question and hypothesis (page 2, line 96-99).

This is a narrative literature review to critically analyse the qualitative findings from the literature, therefore we hope the reviewer would appreciate the differences in research questions and hypotheses compared with a quantitative study.

The research question and hypotheses must then be taken up again in the discussion and addressed with the available literature.

 This requires a fundamental revision of the paper.

We have addressed the reviewer’s comments in the manuscript on Page 9, Lines 356 -357 and 359 -360.

The rest of the Discussion section (Page 10 and 11) went through each theme in details with critical analysis and summary of the evidence to answer the research question.

The authors should add a table with the most important findings of the review.

We appreciate the reviewer’s comment, and we believe that the most important findings according to key themes and sub-themes and their specific issues, are presented in Table 2.

To combine this feedback and the feedback from Reviewer 2, we now present the content of Table 2 in three tables: Table 2a, 2b and 2c.

Strategic and operational recommendations for action would be important to tackle the problem. (Conclusion)

Thank you for this feedback, and we appreciate it.

We added the Strategic and operational recommendations for action in the Conclusion, Page 12, lines 485 -488.

The problems are also transferable to other health systems in Europe due to the explosive immigration of Africans and Syrians.

We appreciate this constructive feedback and add a section to cover this content on page 12, lines 473 -478.

Reviewer 2 Report

Line l26-133 - Authors jumped from second to fourth, skipping third.

There should be a short introduction of each table before the table is illustrated and a follow-up of findings (summary) after each table.

When presenting the various themes mentioned in the tables within the narrative, it might be good to include the section of the table as preliminary to the explanations before each narrative.  Allows quick reference.  Maybe assign a different label to the section off the table. Each theme could be a separate table.  Might be less confusing.  For example, there are three themes in Table 2, so break each our into separate tables (Table 2-1, Table 2-2, Table 2-3).  Or, make each theme a separate numbered table.

Table 1 needs to be less congested.  Modify the layout or make separate tables.

Author Response

Reviewers’ Comments

Authors’ Responses

Line l26-133 - Authors jumped from second to fourth, skipping third.

We have added the missing word, ‘Third’ and edited the sentence on page 3, line 152.

There should be a short introduction of each table before the table is illustrated and a follow-up of findings (summary) after each table.

We have added a short introduction for Table 1 on the page 4, line 167.

A short introduction for Table 2 is presented on page 5, paragraph 1 in the original manuscript.

When presenting the various themes mentioned in the tables within the narrative, it might be good to include the section of the table as preliminary to the explanations before each narrative.  Allows quick reference.  Maybe assign a different label to the section off the table. Each theme could be a separate table.  Might be less confusing.  For example, there are three themes in Table 2, so break each our into separate tables (Table 2-1, Table 2-2, Table 2-3).  Or, make each theme a separate numbered table.

We appreciate the reviewer’s comment and have presented the table according to the 3 themes as Table 2a, 2b, and 2c. Each Table has been cited in the text accordingly.

Table 1 needs to be less congested.  Modify the layout or make separate tables.

The table will be presented in a landscape format in the final print, a change from the current portrait format, which will be much less congested and easier to read.

Reviewer 3 Report

This article is very significant for the health professionals, because the authors, performing a narrative literature review emphasizes the idea that the subjective experience of cancer patients is as important as their clinical characteristics.

1.      By exploring Indigenous people's experiences of cancer care, the authors provide valuable insights and recommendations for improving counseling for ethnic minority patients.

2.      The authors listed 19 keywords. I would suggest reducing their number.

Author Response

Reviewers’ Comments

Authors’ Responses

This article is very significant for health professionals, because the authors, performing a narrative literature review emphasizes the idea that the subjective experience of cancer patients is as important as their clinical characteristics.

1.      By exploring Indigenous people's experiences of cancer care, the authors provide valuable insights and recommendations for improving counseling for ethnic minority patients.

2.      The authors listed 19 keywords. I would suggest reducing their number.

Thank you for your time and effort to review our manuscript and we appreciate your feedback.

The number of the key words is reduced from 19 to 10.

Round 2

Reviewer 1 Report

The authors have implemented the instructions.

For the future qualitative research work of the authors, it is also important to formulate research questions. These are the basis for the research results and discussion.